# Potent pollen gene regulation by DNA glycosylases in maize

Yibing Zeng[1,4], Julian Somers[1,4], Harrison S. Bell[2], Zuzana Vejlupkova [2], R. Kelly Dawe [1,3], John E. Fowler[2], Brad Nelms[3] ✉ & Jonathan I. Gent [3] ✉

Although DNA methylation primarily represses TEs, it also represses select genes that are methylated in plant body tissues but demethylated by DNA glycosylases (DNGs) in endosperm or pollen. Either one of two DNGs, MATERNAL DEREPRESSION OF R1 (MDR1) or DNG102, is essential for pollen viability in maize. Using single-pollen mRNA sequencing on pollen-segregating mutations in both genes, we identify 58 candidate DNG target genes that account for 11.1% of the wild-type transcriptome but are silent or barely detectable in other tissues. They are unusual in their tendency to lack introns but even more so in their TE-like methylation (teM) in coding DNA. The majority have predicted functions in cell wall modification, and they likely support the rapid tip growth characteristic of pollen tubes. These results suggest a critical role for DNA methylation and demethylation in regulating maize genes with the potential for extremely high expression in pollen but constitutive silencing elsewhere.

In angiosperms, a single pollen grain is made up of a pollen vegetative cell that encapsulates two sperm cells. After release from the anther and contact with a stigma, the pollen vegetative cell germinates into a pollen tube that rapidly elongates through the transmitting tract until it reaches the ovary and delivers one sperm to the egg cell to form the zygote and one to the central cell to form the endosperm[1]. Like unicellular root hairs, moss protonema, and fungal hyphae, the pollen tube elongates by tip growth. Also like fungal hyphae, it grows invasively, that is through cell walls and extracellular matrices[2]. In doing so, it secretes proteins that loosen or modify cell walls, including expansins, pectinases, pectin methylesterases, and rapid alkalinization factors (RALFs)[3]. In theory, these factors could act on the tube tip, on stigmatic epidermal cells, or on the extracellular matrix within the transmitting tract. Development of the pollen grain itself involves a complex process of building a multilayered cell wall in coordination with the surrounding tapetal cells[4]. In maize, the pollen tube is among the fastest-growing eukaryotic cells and can reach a rate of 1 cm/h as it travels through a style (silk) that can be 30 cm long[5,6]. In comparison, the rate of fast-growing hyphae is on the order of 1.3 mm/h[7].

The extreme growth rate of the pollen tube raises the possibility that the pollen transcriptome would be highly specialized. Indeed, pollen transcriptomes differ considerably from the transcriptomes of other plant tissues, appearing removed from others in multi-tissue analyses[8]. While some transposable elements (TEs) are known to have enriched mRNA expression in pollen[9–11], repression of TEs is generally maintained in pollen in spite of increased chromatin accessibility because of multiple overlapping mechanisms of repression[12,13]. Some TEs are also hypothesized to be transcribed in the pollen vegetative nucleus in order to produce siRNAs that are transmitted to sperm nuclei to reinforce repression in the next generation[10,11,14,15]. Consistent with robust TE repression in pollen, both sperm and vegetative nuclei have similar or higher DNA methylation levels than other cell types in Arabidopsis[16]. There are notable locus-specific differences though, where the vegetative nucleus is demethylated relative to sperm. DNA demethylation occurs passively, through DNA replication, or actively, by specific replacement of methylated cytosines with unmethylated cytosines.

In plants, active DNA demethylation is accomplished by DNA glycosylases (DNGs), of the same type that function in base excision

[1]Department of Genetics, University of Georgia, Athens, GA, USA. [2]Department of Botany and Plant Pathology, Oregon State University, Corvallis, OR, USA. [3]Department of Plant Biology, University of Georgia, Athens, GA, USA. [4]These authors contributed equally: Yibing Zeng, Julian Somers. ✉e-mail: nelms@uga.edu; gent@uga.edu

repair[17]. These enzymes are essential for endosperm development in diverse angiosperms including Arabidopsis, rice, and maize, where they demethylate maternally imprinted genes (initiating demethylation in the central cell before fertilization[18,19]). In addition, they demethylate thousands of other loci, most of which do not overlap genes at all[10,20–22]. The same DNGs that demethylate DNA in the central cell and endosperm also demethylate overlapping sets of thousands of loci in the pollen vegetative nucleus, as evidenced by comparisons of wild-type and mutant methylomes[10,15,23]. In Arabidopsis, mutants of the DNG DEMETER have a weak and background-specific defect in pollen tube growth, but double mutants lacking DEMETER and another DNG, ROS1, have a stronger phenotype in which pollen tubes growth is disoriented[24]. A DNG in rice, called DNG702 or ROS1A, is also important for pollen fertility, and its mutant has earlier defects in microspore morphology[25,26]. In Arabidopsis, 27 genes have been identified that are demethylated by DNGs in pollen and transcriptionally activated. Likely consistent with the disoriented pollen tube growth in DNG mutants, these genes are strongly enriched for kinases predicted to be involved in protein signaling[24,27]. Maize has four DNGs in three subtypes[20]. The subtype that is highly expressed in endosperm has two paralogous genes, *mdr1* (also known as *dng101*[28] and *zmros1b*[22]) and *dng102* (also known as *zmros1d*[22]). Mutations in both *mdr1* and *dng102* can be transmitted through the maternal gametophyte simultaneously, but the resulting seeds abort early in development[20]. Maternal transmission of either single mutant produces healthy seeds, as does paternal transmission. They cannot both be paternally transmitted together, however, indicating an essential function of these DNGs in the male gametophyte.

Understanding functions for DNA methylation in gene regulation in plants has proven difficult. Part of the difficulty is due to the complexity of DNA methylation and part to the complexity of genes themselves. For example, methylation that silences TEs located in introns has different effects than methylation located in cis-regulatory elements. In cases where regulatory regions contain TEs or tandem repeats, it is difficult to distinguish genome defense mechanisms from normal developmentally or environmentally responsive gene regulation. The Arabidopsis genes *FWA* and *SDC* and the maize genes *b1* and *r1* provide a few of the many examples where gene regulation can be strongly affected by TE-related DNA methylation due to TEs or other repeats in their cis-regulatory elements[29–32]. Exons are frequently methylated in the CG context alone, referred to as gene body methylation (gbM)[33]. This is a common feature of broadly expressed genes, including in the cells where they are highly transcribed. TE-like methylation (teM), where CG methylation (mCG) and CHG methylation (mCHG) together, can also occur in exons and is associated with transcriptional repression. CHH methylation, which is associated with RNA-directed DNA methylation in maize, can also be grouped under teM but is negligible compared with mCHG.

In a recent survey of methylation patterns in maize genes, we identified a large set of genes with teM, as defined by the methylation of coding DNA in the leaf. These genes make up more than 10% of gene annotations across diverse maize genomes[34]. Closer inspection revealed that the vast majority are poorly expressed, not conserved even between cultivated maize stocks, and frequently overlap TE annotations. Intriguingly, a subset of the remaining genes with teM in the leaf is highly expressed in the endosperm or in anthers and tassels[34]. Since tassels and anthers contain pollen, the subset of genes with both teM in the leaf and high expression in anther are good candidates for function in pollen, dependent on developmentally specific demethylation. Together with the requirement for *mdr1* and *dng102* in pollen fertility, these observations led us to explore relationships between DNGs, genes that are repressed by teM in the plant body (sporophyte), and pollen development.

## Results

### High expression of genes with TE-like methylation in pollen

To better quantify the number of expressed, non-TE genes with teM, we enriched for high-confidence gene annotations by including only those encoded at syntenic positions in B73 and 25 other diverse maize genomes that are founders for the nested association mapping (NAM) population (i.e., part of the defined core gene set[35]) and whose coding DNA sequence (CDS) did not overlap with TE annotations. For each of the ten diverse tissues assessed by RNA-seq in the prior study[35], we counted the number of these high-confidence genes with teM expressed at increasing TPM thresholds (Fig. 1a and Supplementary Fig. 1). The two pollen-containing tissues, anther and tassel, clearly

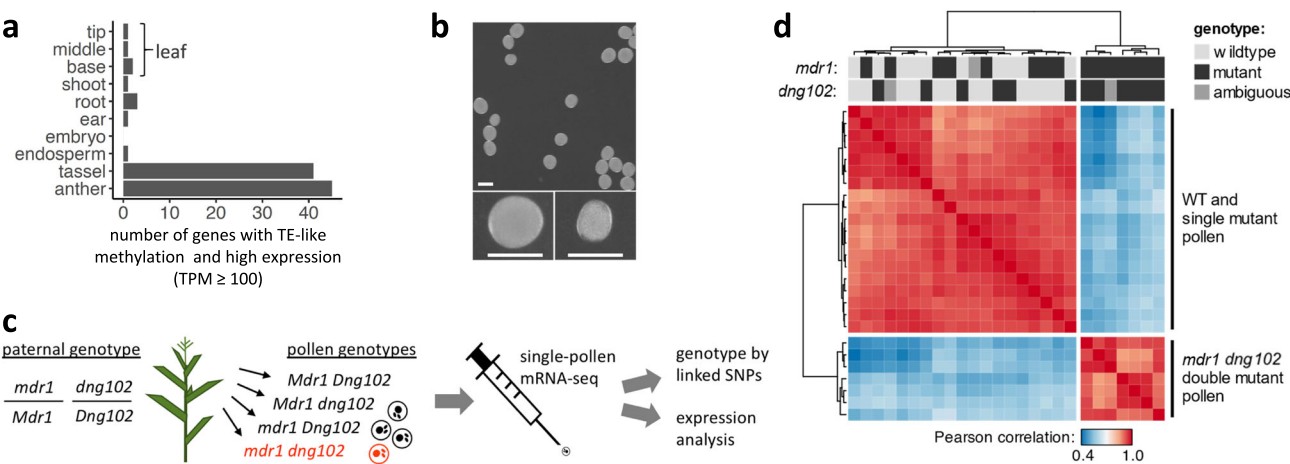

**Fig. 1 | Evidence for DNG function in pollen and summary of experimental approach. a** Expression of genes with teM in anther and tassel. The *X*-axis indicates the number of genes in each of the ten tissues of ref. 34 which have teM and TPM values of at least 100. This analysis only includes high-confidence genes defined as genes that do not overlap with TE annotations and are part of the "core gene" set[35]. **b** Top panel, pollen from an *mdr1 /Mdr1 dng102/Dng102* double heterozygous plant, segregating four haploid genotypes. The bottom panels show large and small pollen grains from a representative plant. Pollen from all six double heterozygous plants imaged exhibited a small pollen (sp) phenotype (Supplementary

Fig. 2). Size bar = 100 μm. **c** Schematic of single-pollen mRNA-seq method. Individual libraries were prepared and sequenced for each pollen grain. Capital indicates WT allele, lowercase mutant, red double mutant. **d** Unsupervised clustering of single-pollen transcriptomes based on Pearson correlation across the entire dataset (all by all). Warmer colors indicate stronger correlations between transcriptomes. The top two rows indicate *mdr1* and *dng102* genotypes inferred from SNPs linked to the two loci derived from RNA-seq data, which were scored independently of the transcriptome correlation analysis. Genotypes: black is mutant, light gray is wild-type, and dark gray is ambiguous.

stood out. With a moderate threshold (TPM ≥ 100), 45 genes with teM were expressed in anther and 41 in tassel (37 overlapped). No other tissue had more than three expressed genes with TPMs of at least 100, and none overlapped with the 45 in anther.

**A small pollen (sp) phenotype associated with *mdr1* and *dng102***
These results pointed toward active pollen expression of genes with teM in the plant body, potentially targeted by DNGs for demethylation. We established previously that *mdr1* and *dng102* loss-of-function mutants could not be transmitted together through pollen (<0.005% transmission[20]), indicating that these two genes are redundantly essential in pollen. To evaluate the function of MDR1 and DNG102, we visually examined pollen from plants that were heterozygous for both *mdr1* and *dng102* mutations to assess whether any strong morphological defect would show up in ¼ of the haploid pollen (Fig. 1b, c). Although there was no conspicuous increase in qualitative morphological defects in these populations of pollen, quantitative analysis of two-dimensional pollen area from microscope images revealed a bimodal distribution of pollen from double mutant but not from single mutant or wild-type plants, i.e., segregation of sp phenotype (Supplementary Fig. 2A). The size of the secondary peak of pollen areas was consistent with an sp phenotype in ¼ of the pollen and a ~35% reduction in area, corresponding to a ~50% reduction in volume. Moreover, the sp phenotype was also present in pollen populations derived from plants carrying a second, independent *mdr1* allele alongside the *dng102* mutant, co-segregating in sibling plants with the parental double mutant heterozygous genotype (Supplementary Fig. 2B).

**Single-pollen RNA sequencing of *mdr1 dng102* double mutants**
We next sought to determine how gene expression was affected in the *mdr1 dng102* double mutant pollen. The inability to generate plants homozygous for both *mdr1* and *dng102* makes traditional bulk expression analysis infeasible. We overcame this by directly sequencing RNA from individual pollen grains[36,37], making it possible to compare double mutant to single mutants and wild-type sibling pollen grains (Fig. 1c). We obtained transcriptomes from 26 individual pollen grains collected from an *mdr1/Mdr1 dng102/Dng102* double heterozygous plant, detecting a mean of 549,559 mRNA transcripts (Unique Molecular Identifiers; UMIs) and 9396 expressed genes per pollen grain.

To determine the individual pollen genotypes, we reasoned that expressed SNPs linked to *mdr1* and *dng102* would allow us to infer the genotype of each pollen grain directly from its transcriptome. Mutant alleles for both genes were originally isolated in a B73 stock but then back-crossed into W22 for five generations[20]; as a result, these plants were predominantly W22 but had several Mb regions of the B73 sequence surrounding each mutant allele. We analyzed SNPs in the single pollen RNA-seq data to determine whether transcripts for genes neighboring *mdr1* and *dng102* were from the W22 or B73 alleles (*mdr1* and *dng102* were expressed at too low a level to genotype directly). Genotypes were assigned for both *mdr1* and *dng102* in 23 of 26 pollen grains (Supplementary Fig. 3); the remaining 3 pollen grains were ambiguous for either *mdr1* or *dng102* due to recombination between the linked genes used for genotyping (Supplementary Fig. 3E, F). In total, we found 4, 7, 6, and 6 pollen grains with the *Mdr1 Dng102*, *mdr1 Dng102*, *Mdr1 dng102*, and *mdr1 dng102* genotypes, respectively, consistent with expectations for random segregation of both mutant alleles (*p* = 0.843; chi-squared test).

**Identification of DNG targets by differential gene expression**
Unsupervised hierarchical clustering of the single-pollen transcriptome data, including the three pollen transcriptomes with ambiguous genotypes, produced two distinct clusters, one with 19 pollen grains and one with 7 (Fig. 1d). These clusters perfectly separated pollen with the double mutant *mdr1 dng102* genotype from all

others (Fig. 1d), showing that there was a strong and reproducible gene expression change in the double mutant. In contrast, there was no separation of *mdr1* or *dng102* single mutant pollen grains from wild-type or from each other, indicating relatively less transcriptional change in the single mutants. To identify genes that were mis-expressed in the double mutant pollen, we used DESeq2 to assess differential expression relative to wild-type and single mutant pollen grains. One hundred and six genes were differentially expressed with moderate cutoffs (adjusted *p*-value ≤ 0.05; ≥ 2-fold change in expression), with 58 exceeding very strong criteria for differential expression (≥8-fold change in expression; mean UMIs ≥ 10). All 58 of these strongly differentially expressed genes (DEGs) were downregulated in the double mutant pollen, with a median decrease of 124.1-fold (Fig. 2a). The 58 DEGs made up 11.1% of all detected mRNA transcripts in WT pollen (Fig. 2b), representing some of the most highly expressed genes. In contrast, these genes made up only 0.3% of transcripts in the double mutant. This is consistent with a model where MDR1 and DNG102 are required to demethylate a set of strong pollen-expressed genes so that they can be properly expressed. There was also a mild reduction in the expression of the DEGs in both *mdr1* and *dng102* single mutant pollen grains, suggesting a slight expression defect in the single mutants; however, these changes were too weak to detect without knowledge of the potential target genes (Fig. 2b and Supplementary Fig. 4), and so it is unsurprising that both single mutants can be readily transmitted through pollen while the double mutant cannot.

**Characteristics of DEGs**
Many of the DEGs shared similar genomic features. Half (28 genes) occurred in six clusters of two to eight differentially expressed copies (Supplementary Data 1 and Fig. 2c, d). The clusters were not simple head-to-tail gene arrays, but included variable amounts of DNA between genes. Some of the clusters also carried additional gene copies that were not detected as DEGs. There was also a strong trend for the DEGs to have only one or two exons. Of the genes in clusters, 27 were annotated with a single exon in the canonical transcript and one with two exons. Of the other 30 DEGs, ten have single exons and five have two exons. Four of the six clusters encode expansins (one alpha type, four beta types), and one encodes polygalacturonases (pectinases). When secreted into the apoplast, expansins, and pectinases loosen cell walls[38,39]. The sixth cluster encodes two uncharacterized proteins of 69 and 76 amino acids with homologs across the grass family (Supplementary Fig. 5). The closest matches to these proteins in Arabidopsis are the arabinogalactan protein AGP11 and its homolog APG6, with the two maize AGP-like proteins showing 23% aa identity match to the first 74 aa of the 136-aa of AGP11. Although the molecular functions of AGP11 and AGP6 are unknown, they localize to cell walls, and inhibiting their function is associated with defects in the nexine layer of the pollen grain wall and in pollen tube growth[40,41]. The 30 singleton DEGs include two beta expansins, two polygalacturonases, one pectin methylesterase inhibitor, and one pectin methylesterase. The pectin methylesterase is part of another cluster of multicopy genes that has a role in overcoming maternal barriers to fertilization as part of the Ga2 unilateral cross incompatibility system[42]. The other three gene copies in the Ga2 cluster were not detected as DEGs. The highest expressed of the DEGs were two more AGP-like unlinked paralogs encoding 70-aa and 72-aa proteins with about 26% aa identity with the two clustered AGP-like DEGs and about 29% aa identity with the first 74-aa of the Arabidopsis AGP11 protein (Supplementary Fig. 5). Two other DEGs were paralogs encoding vesicle-associated membrane proteins, one of which, VAMP726, has been shown to influence lignin content in the maize pollen cell wall[43]. In total, 38 of the 58 DEGs are predicted to have cell wall-related functions, and 32 of those are expansins and proteins related to pectin degradation or modification, likely involved with pollen tube growth. Another gene with potential function in rapid growth is an actin-binding villin protein.

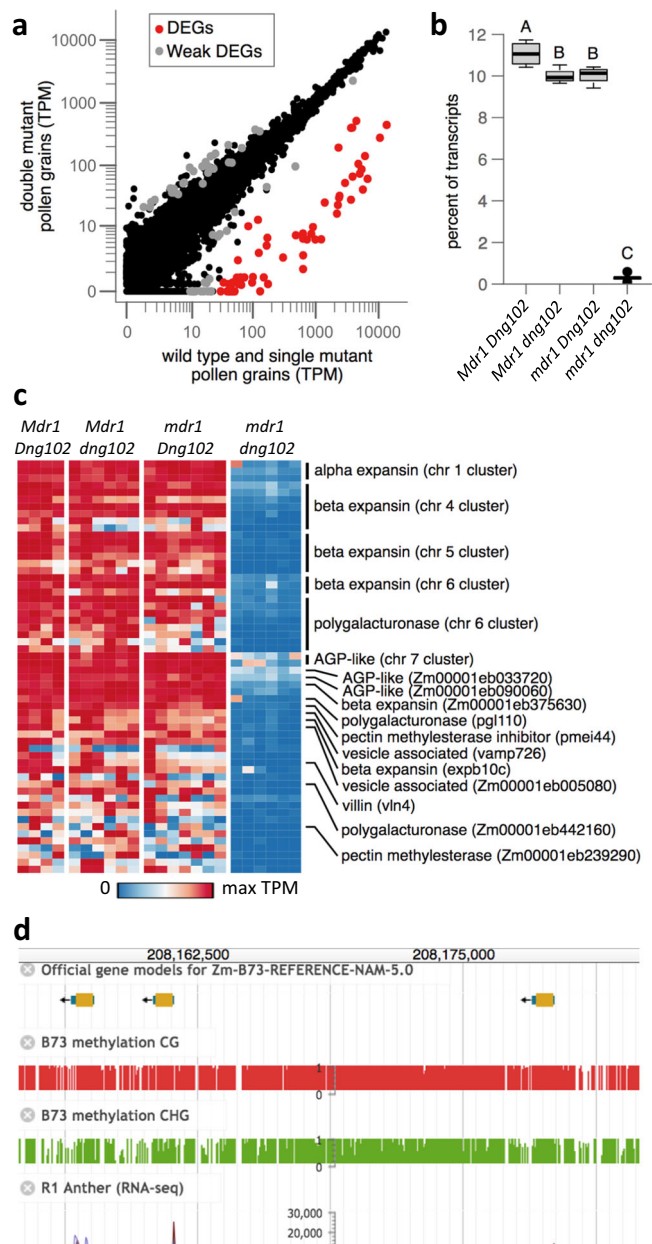

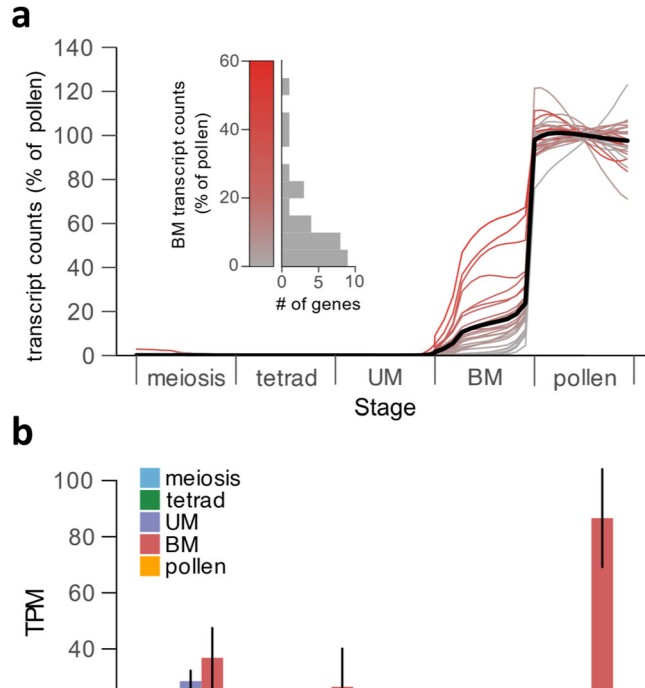

**Fig. 2 | Identification of candidate DNG target genes. a** WT and single mutant vs double mutant gene expression. Dots correspond to individual genes. Axes indicate mean TPM values for each set of transcriptomes. The strong DEGs (red dots) have ≥8-fold change in expression in double mutant and an average of ≥10 UMIs in the WT and single mutant. An additional 48 genes (weak DEGs, gray dots) showed evidence of differential expression by less stringent criteria (adjusted *p*-value ≤ 0.05; ≥2-fold change in expression). Raw (unadjusted) *p*-values were calculated using DESeq2 and then adjusted for multiple hypothesis testing using Holm's method. **b** Total expression of the 58 DEGs as a percent of all measured transcripts. Boxplots indicate the median (horizontal dark line), interquartile range (box), and range (vertical lines) of the measured values. Letters indicate statistical significance: groups not sharing a letter have a significantly different mean (*p* ≤ 0.05; Tukey's honest significant difference test). See "Statistics and reproducibility" in the "Methods" section for individual *p*-values. **c** Expression patterns of DEGs in each pollen grain. Each row represents a single gene, and each column a pollen grain, organized by genotype. Rows are sorted by chromosome, position, and TPM, with genes in clusters listed above singletons. **d** MaizeGDB browser image of an approximately 30 Kb part of a beta expansin gene cluster on chromosome 5. Included are publicly available DNA methylation tracks and anther gene expression tracks[35].

**Fig. 3 | Expression time course of DNGs and targets. a** Expression timecourse of DEGs. The bold line represents the average expression profile across all DEGs. UM unicellular microspore, BM bicellular microspore. Data are from B73/A188 hybrids, normalized to the mean transcript abundance in pollen[36]. The time course spans 349 pollen grains and precursors, here reported as a rolling weighted average by pollen precursor stage (see "Methods"); a heatmap without averaging is visible in Supplementary Fig. 6. The variation seen in mature pollen for two genes in particular is consistent with random noise and not statistically significant. Gray to red color scale indicates the expression level at the bicellular stage, as quantified in the inset. **b** Expression timecourse of the four maize DNGs *mdr1, dng102, dng103*, and *dng105* using the same data as in (**a**). Low expression of these genes makes them unsuitable for the graphical representation used in (**a**). Instead, bar heights indicate average TPM values from individual pollen and pollen precursors, and error bars indicate standard errors. *N* = 119, 22, 175, 11, and 15 single cells or pollen grains for meiosis, tetrad, UM, BM, and pollen, respectively.

The remaining 17 DEGs did not show any clear trends in terms of predicted function.

## Gene expression time course in pollen precursors

To determine when the DEGs were first expressed during pollen development, we examined an expression time course that covers the beginning of meiosis through mature pollen[36]. The DEGs showed undetectable or very little expression during meiosis and early haploid stages, but were then up-regulated to varying degrees at pollen mitosis I (Fig. 3a and Supplementary Fig. 6) corresponding to the major wave of haploid (gametophyte) genome activation in maize[36]. This coincides with peak expression of *mdr1* and *dng102* (Fig. 3b), but unlike their potential targets, both DNGs were also detectably expressed throughout meiosis and early pollen development. Altogether, this suggests that MDR1 and DNG102 act on their target genes sometime before or shortly after pollen mitosis I. While the earliest-expressed DEGs were upregulated at pollen mitosis I, most were not strongly expressed until the mature pollen stage. This might suggest a second wave of MDR1/DNG102 activity, but could also be explained by a single, earlier period of DNA demethylation followed by a later increase in transcription.

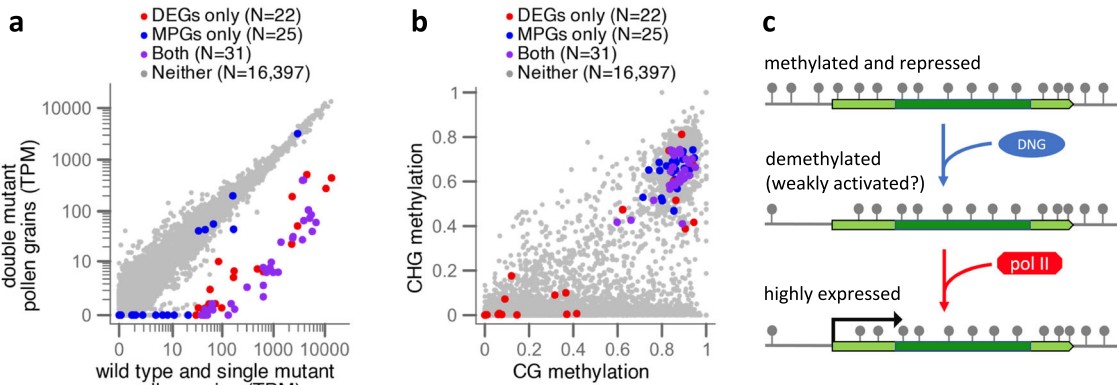

**Fig. 4 | Synthesis of methylation and transcriptome-based results. a** The teM character mapped onto the differential expression analysis comparing WT and single mutant vs double mutant gene expression. Axes indicate the mean TPM values for each set of transcriptomes. MPGs have teM and at least tenfold higher expression in anther than in eight vegetative tissues in B73. Only core genes that are annotated in the W22 genome and in all 26 of the NAM founder genomes and which have sufficient coverage of EM-seq reads were included in this analysis. **b** mCG vs mCHG for the same sets of genes displayed in (**a**). Methylation values are measured in CDS only, as a proportion of methylated cytosines to total cytosines, and range from 0 to 1. **c** A model for DNA methylation in pollen gene regulation, requiring an initial licensing step by DNGs removing methylation (gray lollipops), before transcription at high levels via gene-specific activating factors (not shown) recruiting RNA polymerase II.

## Identification of DNG targets by methylation and expression

Since a major motivation for this study was our observation that a set of genes with teM has high expression in pollen-containing tissues, we asked whether that set overlaps with the candidate DNG target genes we identified as DEGs in the *mdr1* and *dng102* mutant pollen. To answer this quantitatively, we identified genes with teM and syntenic conservation in maize that had at least tenfold more expression in anthers than in the eight other non-pollen-containing plant tissues accessed by RNA-seq in the same study. This produced a set of 56 genes, which we refer to as methylated pollen genes (MPGs) for brevity. While we required only a tenfold increase in gene expression relative to each of the other eight tissues, the average fold change for MPGs was actually far greater because MPGs were either undetectable or barely detectable in these tissues. The MPGs were identified with methylation and expression data from a B73 stock yet strongly overlapped (36 of 56) with the independently identified DEGs from a W22 stock (Fig. 4a, b). The genetic differences between W22 and B73 would be expected to reduce the amount of overlap between gene sets. Consistent with this, an additional 12 MPGs were in the same seven clusters as DEGs, and three more were unlinked paralogs of DEGs (Supplementary Data 1). The five MPGs that were neither in clusters nor paralogs of DEGs encoded an RNA binding protein; an oxalidate oxidase; two WEB domain paralogs implicated in actin-mediated plastid movement; and most striking because of its high expression in pollen, *ralf1*, a member of the rapid alkalinization factor family of secreted small proteins that influence the cell wall via interaction with receptors and other apoplastic molecules[3,44].

## Evidence for conservation in rice

Including *ralf1*, 46 of the 56 MPGs are implicated in cell wall functions. These results raise the question of whether cell wall genes are conserved targets of DNGs. The two largest categories we identified, expansins and polygalacturonases, are common grass pollen allergens. In rice, four genomic clusters containing a total of 19 pollen allergen genes have been identified: three clusters of single-exon expansin genes and one cluster of two-exon polygalacturonase genes[45]. Of these 19 genes, 15 have DNA public methylation data from methylC-seq reads in rice leaf[46]. All 15 have teM in their coding DNA like their maize homologs, as expected for conserved pollen-specific activation by DNGs (Supplementary Fig. 7).

## Expression of *dng105* in pollen and precursors

Assuming the stable expression of some MPGs like *ralf1* in *mdr1 dng102* double mutant pollen is biologically meaningful and not a technical artifact of comparing data derived from W22 to B73, a theoretical explanation is that the more distantly related DNG, *dng105*[20], is also expressed in pollen and pollen precursors (Fig. 3b). In the absence of MDR1 and DNG102, DNG105 might act redundantly on a subset of genes to activate gene expression. Conversely, DEGs with little or no teM in their coding DNA (Fig. 4b) could be explained by indirect effects of demethylation of other genes or by demethylation of their cis-regulatory elements rather than coding DNA. Regardless, the clear pattern is that genes with teM in the leaf and high expression in pollen tend to be dependent on the DNGs MDR1 and DNG102.

## Demethylation of DNG targets in pollen

As illustrated in Fig. 4c, these results are consistent with a repressive function of TE-like DNA methylation in cis-regulatory elements in the plant body, which is removed by DNGs during pollen development to license gene expression. An alternative hypothesis is that DNGs could function to activate gene expression in pollen without demethylating DNA, e.g., by binding to methylated cytosines and recruiting other transcriptional activators. Assaying DNA methylation in pollen is problematic because there are two copies of sperm DNA for each copy of vegetative cell DNA. Assuming maize is like rice and Arabidopsis where demethylation is restricted to vegetative cell DNA only[15,47], then one-third of pollen EM-seq reads are expected to show evidence of demethylation. In addition, constraints on mapping short reads limit the accuracy of methylation quantification at multicopy loci. Nonetheless, a subtle decrease in mCG of MPGs was detectable in EM-seq libraries prepared from whole W22 pollen, especially in promoters (Fig. 5a). As an alternative way to quantify differences in mCG between tissues, we measured the average mCG value of the whole gene for each MPG and its 600-bp upstream region (promoter) (Supplementary Fig. 8). The mean per gene mCG value decreased 7% in pollen relative to embryo. For promoters, it decreased by 17% relative to embryos. While these decreases were small, they were statistically significant ($p = 0.0007$ for genes, $p = 0.001$ for promoters; one-tailed Wilcoxon signed-rank test).

mCHG appeared unaffected in MPGs in pollen. However, the baseline mCHG level in the rest of the genome was elevated in pollen relative to embryo (Fig. 5b). Given the context of a higher mCHG baseline in pollen relative to embryo, the similar mCHG level in MPGs

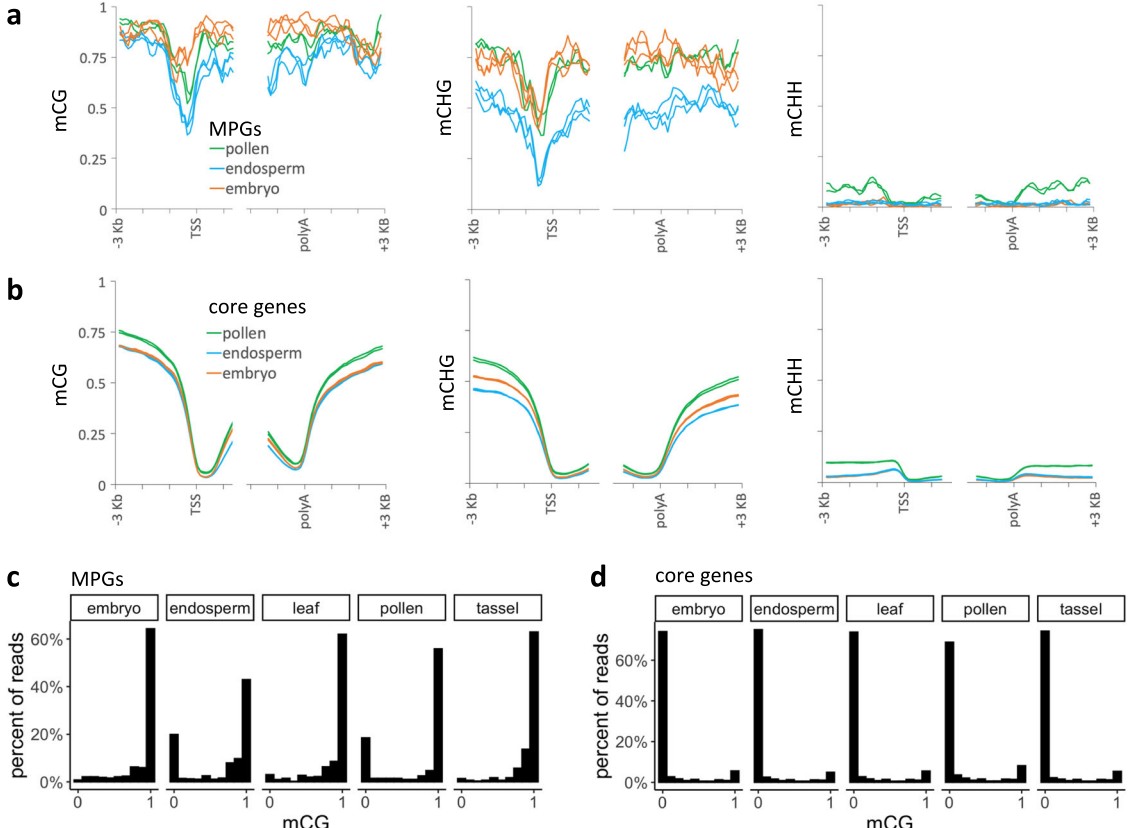

**Fig. 5 | DNA demethylation in pollen and endosperm. a** Metagene methylation profile for MPGs in the W22 genome. These are the 44 of 56 MPGs present in both the W22 and B73 genome annotations. Profiles are centered on transcription start sites and polyadenylation sites (polyA). Values are derived from 100-bp intervals, but the curves were smoothed using moving averages over three 100-bp intervals. Each biological replicate is shown separately (two for pollen, three for endosperm and embryo). **b** Same as (**a**) except all core genes were included in the analysis. Core genes are annotated at syntenic positions in W22, B73, and the 25 other NAM founder genomes. **c** Single-read mCG calls from MPG promoters. Only genes with EM-seq coverage from W22 leaf and mCG values of at least 0.4 in the first 100 bp of their promoters are included. Only pollen and endosperm had a distribution of mCG values that differed significantly from leaf (*p*-value < 0.00005, two-sample Kolmogorov–Smirnov test). **d** Single-read mCG calls from core gene promoters.

could be consistent with partial CHG demethylation limited to one-third of nuclei in pollen. Weaker CHG demethylation compared to CG demethylation could be explained by either a preference of DNGs for mCG or by competing action of CHG methyltransferases adding mCHG in pollen. Likely unrelated to DNGs, mCHH in pollen was unusual in remaining elevated in intergenic regions rather than the usual pattern of rapidly decreasing further from genes.

### Demethylation of DNG targets in endosperm

Several pieces of evidence suggest that *mdr1* and *dng102* may function similarly in pollen and in central cells or endosperm (maternal genome specific). Both genes are highly expressed in pollen and central cells in rice and Arabidopsis, and overlapping sets of loci are demethylated in pollen and endosperm/central cells in rice and Arabidopsis[10,15,23]. To test whether maize pollen DNG targets show evidence for demethylation in the endosperm, we analyzed published EM-seq data from W22 endosperm and compared it with an embryo. Indeed, MPGs exhibited a stronger reduction in mCG in endosperm than in pollen (Fig. 5a), which is consistent with endosperm carrying two demethylated copies (maternally inherited) for each paternal copy (paternally inherited).

### Strong demethylation in a subset of DNA molecules in pollen

A prediction of demethylation being limited to one nucleus in pollen or to maternal DNA in the endosperm is that EM-seq reads should exhibit a bimodal distribution of methylation rather than a gradient of intermediate values. To test this, we took advantage of the fact that each

read originates from a single DNA molecule but can report on multiple cytosines. We counted the number of methylated and unmethylated (UM) CGs for reads that overlapped with the 100 bp regions upstream of MPGs. Since methylation data was from W22 but MPGs were identified in B73, we limited this single-read analysis to gene annotations that were conserved in both genomes and which had EM-seq reads covering promoters and showing evidence for methylation in the W22 leaf. We focused on mCG because it provides the clearest methylation signal (often fully methylated or fully UM). As controls we examined EM-seq reads from three plant body tissues: embryo, developing leaf, and developing tassel. Only pollen and endosperm showed a bimodal distribution of mCG (Fig. 5c). In contrast, the core W22 genes showed a similar distribution of mCG across all five tissues (Fig. 5d).

A limitation of the single-read analysis applied to groups of genes is that it does not rule out the possibility that the bimodal distribution could be explained by some promoters remaining fully methylated and some being fully demethylated in pollen, regardless of whether the EM-seq reads come from sperm or vegetative DNA. To test this, we selected four individual MPGs with read coverage over their promoters for single-read methylation analysis (an AGP-like gene, a beta expansin, a polygalacturonase, and a pectin methylesterase). We extracted all reads and segments of reads that overlapped with the 600 bp upstream regions of each of these genes, and we defined UM reads as ones with no more than one of five CGs per read being methylated. The distribution of mCG for each of these four genes in pollen was consistent with one-third of reads being UM and two-thirds methylated

(Supplementary Fig. 9). In endosperm, the ratio of UM reads was higher, close to two-thirds UM and one-third methylated, which matches the expected ratio for demethylation of the two maternal copies but not of the single paternal copy. One of the three genes, the polygalacturonase, was an exception and appeared fully methylated in the endosperm. For each of the four genes, the control tissues had fewer UM reads than either endosperm or pollen, though the read counts for individual genes were too low for statistical significance. The percent UM reads obtained from combining all four genes together was 33% for pollen, 47% for endosperm, 1% for embryo, 9% for leaf, and 6% for tassel. For embryo, leaf, and tassel, these numbers are significantly lower than 33% ($p$-value $< 0.005$, one-tailed, one-proportion $z$-test). Since we defined MPGs in part by tenfold higher expression in anther than in endosperm, the fact that at least some of the same loci are demethylated in endosperm also indicates that DNA demethylation alone is not sufficient for high expression, and other factors that are lacking in endosperm are also required for their expression in pollen (Fig. 4c).

## Discussion

Two different approaches converged on similar sets of genes—especially genes that are highly expressed in pollen, encoded in one or two exons and are predicted to modify cell walls. One approach identified genes based on differential gene expression in DNA glycosylase mutants known to be essential for pollen function. The other identified genes are based on TE-like DNA methylation in the leaf and on anther-specific gene expression. Combined with prior evidence for DNGs in the pollen vegetative nucleus driving gene expression needed for pollen fertility in Arabidopsis and rice, these results indicate that DNG-mediated gene regulation in pollen is widely conserved in angiosperms. While the methylation of expansins and polygalacturonase gene clusters in rice leaf suggests similar genes are activated by DNGs as in maize, the data from Arabidopsis DNG mutants suggest an enrichment for genes involved in cell signaling controlling orientation of growth[24,27]. An earlier study in Arabidopsis, however, also noted that pollen-expressed genes with teM (in leaf and mixed-stage inflorescence) were enriched for functions in cell walls[48]. (That study used the term "RdDM targets" as the term "teM" had not been adopted yet). These differences in target genes between the two species may explain differences in phenotypic effects caused by loss of DNG function—complete pollen infertility in maize vs a mildly reduced transmission (with aberrant pollen tube growth orientation and some reduced pollen tube germination) in Arabidopsis[24,49]. In maize, a primarily outcrossing species with an extensive stigma for pollen reception and pollen in excess, rapid pollen tube germination and growth are critical for successful competition and eventual fertilization. Although the expansins, pectinases, and pectin methylesterases regulated by DNGs in maize pollen predict a pollen tube growth defect in *dng* mutants, an earlier pollen defect might prevent a tube phenotype from ever manifesting. In theory, the lack of a single DNG target could lead to a sp phenotype with a large impact on pollen fertility, similar to *sp1*, *sp2*, and *stt1* mutants[50–52].

Maize DNGs, including *mdr1* and *dng102*, are expressed in other cell types, particularly endosperm. There are notable differences between the DNG target genes we identified in pollen and the ones already identified in the endosperm. In diverse angiosperms, key endosperm genes that are upregulated by DNGs (specifically from the maternal genome), function in gene regulation themselves, producing indirect effects on gene expression. These genes include members of the polycomb repressive complex PRC2 and ethylene signaling pathways that have central roles in early endosperm development[53–57]. Thus, moderate expression of these genes initiates a cascade of gene expression changes indirectly resulting from DNA demethylation. In contrast, the DNG target genes in pollen appear to be massively upregulated directly. In endosperm, DNG target genes exhibit a strong tendency for methylation in their promoters and 5′ UTRs rather than CDS[20], whereas the pollen genes are methylated not just in promoters and 5′ UTRs but also across CDS. In fact, our observation of methylation in CDS partially motivated this study[34].

In some animals, DNA methylation in promoters functions with other chromatin modifications as a developmentally stable form of transcriptional repression. This occurs in the repression of germline genes in somatic cell lineages and across the X chromosome in X inactivation[58]. Although there are many examples of repetitive elements acting as cis-regulatory elements that sensitize gene expression to DNA methylation, such as the Arabidopsis *FWA* and *SDC* genes and the maize *r1* and *b1* genes[29–32], DNA methylation is not a common part of developmental gene regulation in plants. Rather, as well established in maize, cis-regulatory elements remain constitutively free of methylation, regardless of which cells the genes are expressed in[35,59]. This is also true of their coding DNA methylation, except in CG context[34]. Keeping cis-regulatory elements free of methylation may be a major function of DNGs in the plant body, allowing dynamic access of both activating and repressing factors[60]. Transcription factors provide both tissue specificity and sequence specificity to repression by recruiting histone modifiers such as the polycomb repressive complexes PRC1 and PRC2, which ubiquitinate histone H2A at lysine 119 (H2AK119ub) and methylate histone H3 lysine 27 (H3K27me3)[61,62].

The vast majority of genes in the pollen vegetative nucleus are likely regulated using the same mechanisms as other plant cells since they are neither differentially expressed in *mdr1 dng102* double mutant pollen nor have teM in the plant body. We hypothesize that the highly expressed DNG target genes we identified require a two-step activation in pollen—first, recruitment of DNGs to create an environment permissive for RNA polymerase, followed by recruitment of other activators and high levels of RNA polymerase itself (Fig. 4c). As in endosperm, unidentified factors would recruit DNGs to some methylated loci and not others. Transcription factors, because of their roles in guiding protein complexes to specific loci would be good candidates. This two-step activation using DNA methylation as the basis for repression could allow for the huge dynamic range of expression we observe, from nearly undetectable in most cells to 11% of the transcriptome in pollen.

Why not use DNA methylation in gene regulation more broadly? One possibility is that the unique epigenetic features of the pollen vegetative nucleus make it better suited to this form of gene regulation[12,13]. Another reason for limiting the use of DNA demethylation in gene activation could be an elevated risk of mutation associated with excising methylated cytosines. Regardless, these results point to a role for DNA demethylation in potent and cell type-specific gene regulation in the pollen vegetative nucleus. This form of regulation not only allows for massive upregulation of gene expression in pollen, but also for robust silencing outside of pollen.

## Methods

### Pollen phenotyping

100 to 600 uL of freshly shed pollen was mixed with 800 uL EAA fixative solution (3:1 ethanol to acetic acid) by inverting three times in a microcentrifuge tube before being parafilmed and stored at 20 °C. Samples were imaged using a LEICA M205 FCA Fluorescence stereo microscope and the Leica Application Suite X (V 3.7.5.24914). Resulting images were imported into Fiji (ImageJ) (V 2.0.0) and subjected to the following processing pipeline: 'Image -> Enhance Contrast' (0.3%), 'Adjust -> Threshold Image' (Auto -> Apply), 'Process -> Binary -> Make Binary',' Process -> Binary -> Watershed', 'Analyze -> Analyze Particles (size (micron²): 2000–14,000, circularity: 0.75–1.00, show: nothing, display results, summarize, exclude on edges, include holes) -> Okay'. Particle measurements were copied into a.csv file and loaded into R (V. 4.2.3) to produce plots with ggplot2 (V 3.4.3) and ggridges (0.5.4) packages.

## Identification of MPGs from B73 expression and methylation data

To identify genes with teM, we reanalyzed methylation data from B73 developing leaf[35] using the same methods as in our recent study of methylation in genes[34]. This method only makes use of methylation within annotated CDS, as introns often have teM for the simple reason that they contain TEs, and UTRs are difficult to annotate accurately. To include more genes with short CDSs, we required only 30 individual informative CGs and individual informative 30 CHGs per gene rather than 40 of each. "Informative" means they were spanned by at least one EM-seq read where the C in the genome could unambiguously be associated with either a C or a T in the read. As before, genes with average methylation levels of at least 40% in both mCG and mCHG were defined as having teM. We only included the core gene set in these analyses, the ones that are present at syntenic positions in B73 and the 25 other NAM founder genomes, as defined previously[35]. This yielded 926 genes with teM, 7882 with gbM, 13,098 that were UM, 3661 that had intermediate methylation values (ambiguous), and 2724 that did not meet the requirements for sufficient informative cytosines. These methylation epialleles are listed along with gene names and methylation values in a table available on GitHub (https://raw.githubusercontent.com/dawelab/MethylatedPollenGenes/main/Data/df_RedefineEpaiele.csv?token=GHSAT0AAAAAACMFLIBUHCBROGBSM7N7KK6AZNKUPWQ).

To more meaningfully quantify the expression of genes with teM in the ten tissues, we further enriched for functional genes by excluding all gene annotations whose CDS overlapped with annotated TEs. Gene and TE annotations were the same as the ones used in the prior study, obtained from https://download.maizegdb.org/Zm-B73-REFERENCE-NAM-5.0/. Using a combination of Unix cut, awk, and sed commands, we converted the source gff3 gene annotation file ZM-B73-REFERENCE-NAM-5.0_Zm00001eb.1.ggf3 into bed format with a geneID for each CDS region. We then used the BEDTools v2.30.0[63] intersect tool to identify all genes whose CDS overlapped with TEs in Zm-B73-REFERENCE-NAM-5.0.TE.gff3 by even a single base. Then we used awk to select the geneID columns and uniq to remove redundant rows corresponding to different CDSs from the same gene. We imported the geneIDs with TE-overlapping CDSs into an R data frame with row names as genes and a second column indicating TE overlaps by a value of 1. Then we merged this data frame with a list of all core B73 genes using the R merge function to create a new data frame where genes whose CDS did not overlap TEs had "NA" in the second column. Finally, we replaced all NA values in the second column with zeros and used this to filter each subset of genes to remove ones with TE-overlapping CDSs using the R tidyverse filter function. This yielded 394 teM genes, 6544 gbM genes, 11,873 UM genes, 3975 that had intermediate values (ambiguous genes), and 2383 that did not meet the requirements for sufficient informative cytosines. These methylation epialleles are also listed along with gene names and methylation values in the table above.

To count the numbers of genes that expressed at or above different TPM thresholds in each tissue, we used the TPM matrix produced in the prior study[34] (https://raw.githubusercontent.com/dawelab/Natural-methylation-epialleles-correlate-with-gene-expression-in-maize/main/Data/B73.all.csv). We used the R merge function to combine data frames containing the methylation epiallele information for the core genes that did not have TE-overlapping CDSs with their TPM values in each tissue. Then we obtained counts of expressed genes in each tissue at each TPM threshold using the R group_by function to process each tissue separately followed by the summarize and sum functions. We then used a for loop in R to iterate over a series of TPM thresholds from 1 to 100.

## AGP-like sequence comparisons

We used NCBI blastp to identify the best homologs of the AGP-like proteins in sorghum, rice, and wheat using default parameters with the "Non-redundant protein sequences (nr)" as Database, "grass family (taxid:4479)" as Organism and sequences of Zm00001eb316010_P001 and Zm00001eb033720_P001 as Query. The resulting GenBank accessions of the best matches were used for sequence comparisons: KAG0544127.1 (Sorghum bicolor), EAZ09485.1 (Oryza sativa), XP_044385131.1 (Triticum aestivum), KAG0524731.1 (Sorghum bicolor), ATS17269.1 (Oryza sativa), and XP_044386294.1 (Triticum aestivum). We used Geneious® 10.1.2 Tree Builder for protein tree construction using global alignment with free end gaps, identity cost matrix, Jukes-Cantor Genetic distance model, UPGMA method, gap open penalty of 6, and gap extension penalty of 3. The three Arabidopsis thaliana arabinogalactan proteins we included in the tree are AT3G01700.1 (AGP11), AT5G14380.1, (AGP6) and AT5G64310.1 (AGP1). For pairwise comparisons between maize proteins and AGP11, we obtained amino acid identities using Geneious® 10.1.2 global alignment with free end gaps, cost matrix identity, gap open penalty of 12, and gap extension penalty of 3. Zm00001eb316010 was 23% identical to AGP11, Zm00001eb033720 was 29% identical to AGP11, and Zm00001eb316010 and Zm00001eb033720 were 26% identical to each other.

## Single-pollen mRNA sequencing

For plant material, the EMS4-06835d allele of mdr1 and the dng102-Q235 allele of dng102 were used[20]. Both alleles originated in B73 and had been backcrossed for five generations into W22 (mdr1 as stock J657 and the dng102 stock as stock J658). Both stocks were then crossed together to create the double heterozygous stock EMS4-06835d/Mdr1 and dng102-q235/Dng102, which was planted in late spring 2022 in a greenhouse in Athens, GA, and grown under ambient light conditions until pollen collection in July.

Single pollen isolation and RNA-seq library prep were performed as described previously[37]. Briefly, pollen was released from anthers into a drop of 0.1× PBS by cutting transversely with a scalpel. Individual pollen grains were then manually isolated with a syringe needle and placed on the cap of a PCR 8-tube strip. RNA-seq libraries were then prepared with CEL-seq[37]. Oligos for first-strand cDNA synthesis are available in Supplementary Data 2 (this replaced the "Barcoded CEL-seq primer plate"); all other oligos and reagents are as described[37].

Sequencing data were analyzed similarly to Nelms and Walbot[36]. Read 2 of these CEL-seq libraries contain ten nucleotides (nt) UMIs, followed by a six nt sample-specific barcode, and then a long string of Ts originally from the mRNA polyA tail. Read 1 contains a sequence matching the mRNA transcript. Paired-end reads were first demultiplexed based on sample-specific barcodes in read 2, requiring a perfect match to one of the expected barcode sequences (Supplementary Data 2). UMIs were then extracted from read 2 and appended to the read 1 sequence identifiers. No more information was used from read 2 and the remainder of the analysis was on read 1 only.

Prior to mapping, reads were trimmed and filtered using Fastp v0.23.2[64] with parameters -y -x −3 -a AAAAAAAAAAAA. Then filtered reads were mapped to the B73 v5 genome using Hisat2 v2.2.1[65] with the parameter -dta. Novel transcripts were annotated and existing ones were extended by de novo transcript assembly using Stringtie v2.2.1[66], guided by the reference gene annotations and using the strand-specific information available from CEL-seq (parameter '-rf'; CEL-seq libraries map specifically to the coding strand). Reads were then assigned to genes using featureCounts v1.2.5[67] with parameters -s 1 -read-Extension5 500. Then unique transcripts were counted using the umi_tools v1.1.2[68] 'count' function with parameter−per-gene. The GTF file of gene annotations and a table of UMI counts for each pollen transcriptome are available in the accompanying source data. For pollen grain genotyping (Supplementary Fig. 3), mapping was performed as above but using the W22 v2 reference genome[69]; the reason for this difference was purely historical: we initially aligned to W22, but found that it was easier to work with B73 annotations because of greater consistency with other datasets.

## Quality control for RNA-seq samples

In total, 48 pollen grains were collected and sequenced. The resulting libraries showed two clear populations with varying library complexity. Twenty-seven pollen grains had high read depth, with a mean of 548,866 UMIs detected per pollen grain (range: 192,456–866,263). The remaining twenty-one pollen grains had much lower read depth, with a mean of 5444 UMIs detected per pollen grain (range: 2969–17,197). These two populations showed no enrichment for a given genotype or sample batch; a likely explanation is that some of the pollen grains failed to lyse completely, resulting in low library complexity. The 21 pollen grains with low complexity were excluded from further analysis.

One additional pollen grain was excluded because it showed several anomalous behaviors. First, it had a relatively low correlation with every other pollen grain in the dataset. Second, when genotyping the genes near *mdr1*, there was a consistent trend for expression from both the B73 and W22 alleles; there was no other sample with noticeable biallelic expression, and this trend was not true for any genes near *dng102*. The conclusions of the paper were not sensitive to the inclusion or exclusion of this one pollen grain, but given the anomalies in the data, it was excluded.

## Analysis of single-pollen RNA-seq data

To genotype individual pollen grains, mapped RNA-seq data were visualized using the integrative genomics viewer (IGV)[70]. Three "sentinel" genes were selected on both sides of both the *mdr1* and *dng102* genes (12 genes in total), based on the availability of mapped RNA transcripts with SNPs that distinguished the B73 vs W22 alleles. The mapped data were visualized to assign each sentinel gene to the B73 or W22 alleles. The B73 alleles are linked to the mutant alleles of *mdr1* and *dng102* while the W22 alleles are linked to wild-type. Some positions were scored as ambiguous if there were no reads spanning the SNPs that distinguish B73 and W22. The sentinel genes were then used to infer the alleles of *mdr1* and *dng102*, requiring consistency in allele calls on both sides (Supplementary Fig. 3).

For the correlation heatmap in Fig. 1d, the expression count matrix was normalized to transcripts per million (TPM) and then log-transformed after adding a pseudocount of 1. Genes with a mean expression under 500 TPM were filtered, and then the pairwise Pearson's correlation was calculated between all samples.

Differential gene expression analysis was performed using DESeq2[71] with default parameters. Unadjusted *p*-values (two-sided) were then adjusted for multiple hypothesis testing using Holm's method. Significant genes were identified as follows: for the "DEG" set, we required an adjusted *p*-value ≤ 0.05, an estimated log2 fold change ≥ 3, and a baseMean ≥ 10; for the "weak DEG" set, we required an adjusted *p*-value ≤ 0.05 and a log2 fold change ≥ 1.

## Time course of gene expression during pollen development

For Fig. 3, the mapped transcript count matrix from Nelms and Walbot (2022) and associated sample metadata (e.g., the developmental stage of each sample) was downloaded from the Gene Expression Omnibus (accession GSE175505). These data, from a B73/A188 hybrid, were mapped to the B73 v4 maize genome[72], and so we determined the v4 IDs of the strong DEGs using the maizeGDB "Translate Gene Model IDs" tool; 41 of 58 DEGs had an associated v4 ID (Supplementary Data 3). We further excluded 12 genes that had very low expression in pollen in the 2022 dataset (<10 TPM), as these may result from incorrect mapping. This left 29 DEGs that were analyzed for their timing during pollen development.

There are large changes in the total number of mRNA transcripts per pollen grain or precursor at different stages of pollen development, and so normalization methods that assume a constant total transcriptome size can be misleading[36]. For example, a gene with the same number of transcripts in a nearly-quiescent cell and in a transcriptionally active cell would highly appear to be differentially

expressed by conventional TPM measurements. Thus, to better determine both the timing and level of DNG target expression, we used a normalization strategy that accounted for the differences in total transcript abundance between pollen and each pollen precursor stage. The data was first normalized to TPM, but then scaled based on the fraction of absolute transcripts detected at a given stage relative to pollen. For instance, a mean of 133,905 and 377,873 UMIs was detected per individual BM stage precursor and mature pollen grain, respectively. Thus, all of the TPM-normalized data for BM stage precursors was scaled by 35.4% (133,905/377,873), preserving the relative difference in total transcripts between BM and pollen. The main effect of this choice on our conclusions is that all DEGs were expressed at a lower level in BM than Pollen (Fig. 3), while if using TPM normalization there were three genes with higher expression in BM than Pollen. Thus, the TPM normalization might lead to the misleading conclusion that these three genes were downregulated between BM and pollen, even though the data is most consistent in a situation where these three genes continue to increase in transcript abundance between BM and pollen, but at a lower rate than many other genes. The proportion of an enzyme's transcripts relative to total transcripts is usually a good indicator of its activity in different cell types regardless of the cell's total transcriptome size. Thus, for measuring DNG transcript abundance in Fig. 3b, we use conventional TPM normalization.

For the time course in Fig. 3a, the normalized transcript abundances were also smoothed using a weighted average to suppress sample-to-sample noise. The average was performed using a Gaussian kernel with the sample-specific *x*-coordinates given as the sample "pseudotime" value as previously defined[36]. This creates a weighted average where two samples that are more similar in overall expression (e.g., similar pseudotime values) are given more weight than samples that are distinct. The effect of this smoothing is similar to taking the mean expression value by stage, but allows more continuous time resolution without requiring sharp stage boundaries (e.g., a gene that goes up within a stage could be recognized). The heatmap in Supplementary Fig. 6 shows the same data without any smoothing

## Statistics and reproducibility

Forty-eight pollen grains were chosen for single-pollen RNA-seq in the experimental design to obtain a minimum of three high-quality transcriptomes of each genotype as biological replicates. Library preparation and initial clustering of single-pollen transcriptomes were done blindly to their genotypes. After validation that genotypes clustered as expected, differential gene expression was done without blinding. Tukey's honest significant difference test was used to test for differences between genotype: $p = 0.001$ for *Mdr1 Dng102* vs *mdr1 Dng102*, $p = 0.001$ for *Mdr1 Dng102* vs *Mdr1 dng102*, $p = 1 \times 10^{-7}$ for *Mdr1 Dng102* vs *mdr1 dng102*, $p = 0.9997$ for *mdr1 Dng102* vs *Mdr1 dng102*, $p = 1 \times 10^{-7}$ for *mdr1 Dng102* vs *mdr1 dng102*, $p = 1 \times 10^{-7}$ for *Mdr1 dng102* vs *mdr1 dng102*.

## Preparation of pollen EM-seq libraries

Fifty to one hundred milligrams of W22 pollen at −80 °C was homogenized in 2-ml tubes using a GenoGrinder (SPEX SamplePrep 2010) with five 3-mm glass beads (Fisher Scientific #11-312A) for 10 min at max frequency (two sets of 5 min with the GenoGrinder rack in each orientation). DNA was extracted using a CTAB extraction buffer containing 1% PVP (*w/v*) and 120 ug/ml proteinase K and purified with chloroform: isoamyl alcohol (24:1) and ethanol precipitation. RNA and degraded DNA were removed by treating 800 ng of DNA (as measured by Qubit) with 1 ul RNase Cocktail Enzyme Mix (ThermoFisher #AM2286) for 30 min at room temperature then size selecting for large molecules with a 0.8:1 ratio of Mag-Bind (omega BIO-TEK #M1378-00). EM-seq libraries were prepared using pollen from two different plants using a NEBNext Enzymatic Methyl-seq Kit (New England Biolabs #E7120S). The input for each library consisted of 100 ng of DNA that

had been combined with 1 pg of control pUC19 DNA and 20 pg of control lambda DNA and sonicated to fragments averaging ~700 bp in length using a Diagenode Bioruptor. The protocol for large insert libraries was followed with formamide as the denaturing agent, and libraries were amplified with 6 PCR cycles and Illumina sequenced using paired-end 150 nt reads.

## Metagene and single-read methylation analyses

In addition to the W22 pollen EM-seq data produced in this study, data from four other W22 tissues were included. These are 15-DAP endosperm and paired embryo (~3.5 mm in width and 5 mm in length), premeiotic tassel (~1 cm stage), and second leaf (prior to emergence from being wrapped in the 1st leaf). See Supplementary Table 1 for SRA accession numbers. EM-seq reads from pollen were trimmed of adapter sequence and mapped to the W22 genome[67] using the same methods as the other tissues[20]. Metagene methylation profiles were produced using the CGmapTools MFG tool v1.2[71]. Methylation values for whole genes and promoters (upstream 600-bp) were obtained with the CGmapTools MTR tool. Genes or promoters lacking at least 30 informative CGs were removed from the analysis. Since the number of informative cytosines varied depending on the source methylome data, each of the five tissues retained different numbers of genes and promoters. For the Wilcoxon signed rank test, only ones that were retained in both embryo and pollen were included. A one-tailed test was used because the known activity of DNGs in removing methylated DNA provided a clear expectation of decreased methylation in pollen. To determine differences in methylation between embryo and pollen for each gene, we excluded genes where mCG equaled zero in either tissue. This lessens noise associated with counting small numbers of EM-seq reads, avoids dividing by zero, and prevents artificially inflating the methylation decrease in pollen. For the single-read analysis of promoter methylation, we were specifically interested in testing whether promoters that are methylated in the plant body are demethylated in pollen. Since MPGs were defined by methylation in their CDS, not promoters, some might lack methylation in their promoters in the plant body and confound the analysis. Thus, we first selected the subset of MPGs with evidence for methylation in promoters in the W22 leaf. In particular, methylation values for the upstream 100 bp of each W22 gene were determined using the CGmapTools MTR tool, and only the subset of 36 MPGs with EM-seq coverage and mCG values of at least 0.4 included in the single-read analysis. All EM-seq reads from each source tissue that overlapped with these regions were selected using the BEDTools v2.30.0[62] intersect tool. A custom Python script, MethylBammerAll.py, was used to call methylation for each read. Only reads with at least four CGs were included. For the single-read analysis of four representative MPG promoters, reads pairs that overlapped specific regions were selected from bam files using the SAMtools view tool v1.17[72] with region parameters as follows: 3:86,636,030–86,636,629, 5:152,375,079–152,375,678, 7:133,679,456–133,680,055, and 9:16,361,514–16,362,113. A custom Python script, MethylBammer.py, was used to trim reads that extended outside these regions and call methylation for each read. Only reads with at least four CGs were included. Methylation calls were visualized using a GGplot2 point geom plot with X-values scattered by a random number generator.

## Reporting summary

Further information on research design is available in the Nature Portfolio Reporting Summary linked to this article.

## Data availability

Raw RNA-seq and EM-seq data generated in this study are available through NCBI BioProject database under accession number PRJNA1035166 [https://www.ncbi.nlm.nih.gov/bioproject/1035166]. Processed single-pollen RNA-seq data (UMI counts) and modified gene annotations used for defining DEGs are available in the Source Data provided with this paper. Processed RNA-seq data and gene methylation data used to define methylated pollen genes are available at https://github.com/dawelab/MethylatedPollenGenes. Source data are provided with this paper.

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

## Acknowledgements

We thank Dong won Kim and Jackson Macdonald for field and greenhouse work and genotyping, and Mary Washburn for sequencing library preparation. This study was supported in part by resources and technical expertise from the Georgia Advanced Computing Resource Center, a partnership between the University of Georgia's Office of the Vice President for Research and the Office of the Vice President for Information Technology. This study was funded by NSF grant 2218712 to J.I.G., J.E.F., and B.N.; NSF grant 204138 to J.E.F.; and NIH grant 1R35GM151237 to B.N.

## Author contributions

Y.Z., J.S., H.S.B., B.N., and J.I.G. analyzed and curated data. H.S.B., Z.V., B.N., and J.I.G. prepared experimental materials and collected data. R.K.D., J.E.F., B.N., and J.I.G. designed the study. R.K.D., J.E.F., B.N., and J.I.G. wrote the paper.

## Competing interests

The authors declare no competing interests.
