## [Peer Review File · Nature Communications]

REVIEWER COMMENTS

Reviewer #1 (Remarks to the Author):

In this manuscript, Zeng and Somers et al performed single pollen RNA-seq using maize mutants heterozygous for two DNA demethylase genes. Because many genes that are normally methylated like TEs in somatic cells highly express in pollen-containing tissues, and double loss of function mutants of DNA demethylase genes show no transmission through pollen to next generation, the authors hypothesized that the TE-like methylated genes are demethylated and activated in pollen and those genes are important for pollen viability and function. The single pollen RNA-seq results clearly support the idea that DNA demethylase genes MDR1 and DNG102 are important for establishing pollen transcriptome. Single pollen RNA-seq results apparently show that double mutant pollens are different from single and WT pollens in overall transcriptome pattern and expression of cell wall related genes that tend to cluster in chromosomes are strongly downregulated. However, link of this downregulation with DNA methylation is not convincingly proved. For example, they can at least analyze DNA methylation in wild type pollen or pollen-containing tissue to see if the target genes (DEGs) are demethylated in pollen or not. In addition they can analyze DNA methylation pattern and/or DNA sequence features that may distinguish the genes that the DNA demethylases work on and genes/TEs that they do not, which may reveal important perspective on how DNA demethylases specifically target genes that should be activated in pollen. In conclusion, in the current form, I think the advancement of the understanding of the function of DNA demethylases in pollen function is limited.

Reviewer #2 (Remarks to the Author):

In this manuscript by Zeng et al, the authors uncover a fascinating class of genes that are highly expressed in maize pollen and are characterized by having TE-like methylation and also being down regulated (fail to switch on) in pollen in DNG mutants. The genes are enriched for functions related to cell wall modification and also dng mutants have pollen defects. This all supports a model whereby DNGs regulate the expression of genes critical for pollen function, presumably by demethylating them in pollen cells. This is a compelling story and points to an important function of the DNGs (although perhaps an indirect function, considering they do demethylate many other loci too), and the use of single pollen RNA-seq was an inspired approach. That said – and as acknowledged by the authors – pollen expression of methylated genes was previously reported in Arabidopsis, although this was investigated in less detail in the prior work. I have no doubt this is an important contribution and will be a highly cited paper in years to come. A couple of questions do come to mind that could make this a more complete discovery.

Major queries:

I would be itching to measure the demethylation of the genes in pollen cells to definitively establish that these loci are indeed demethylated in order to be expressed. This seems highly likely, but that data would be so helpful (I am not sure what the alternative scenario would be – demethylation of regulatory regions or a key TF? – but you never know!). Likely this is a technically challenging thing to do, probably requiring isolating VC nuclei at least in WT pollen. But this could also be discussed (eg in the discussion) as the TE-like methylation profiles are from leaves (?) and I assume a pollen cell methylome is not available for comparison. Or what about looking at methylation in the endosperm (as a proxy for DNG activity in the central cell), given that prior work in Arabidopsis and rice found that many loci demethylated in the vegetative nucleus were overlapped with loci demethylated in the central cell (eg Ibarra et al and Kim et al)?

A second limitation to note is the potential role of DNG105 in pollen. The expression pattern of DNG105 (Fig 3) combined with the stable expression of some of the “pollen methylated genes” in the pollen of the *dng* double mutant, does point to a role for DNG105, although without combining a *dng105* mutant with the other mutations, it is hard to say for sure and we can only speculate. Again, this is not something that can be easily done, and the authors do acknowledge this limitation.

A third query is whether these genes, and the process, is conserved in other species. For example, are the same genes (homologs) TE-methylated and pollen expressed in rice or sorghum? It would be even more compelling to see this in another species, or to determine if this is maize specific.

Minor

- When referring to genes having TE-like methylation, it would be best to specify the tissue the methylation data is from eg “genes with TE-like methylation in leaf tissue”.
- It is mentioned that there were 6 double mutant pollen grains sequenced, but there are 7 samples that cluster with double mutant transcriptome (Fig 1D)? Is it 6 or 7?
- In Fig 2 A and B some axis numbers are cut off.
- Is it known if these pollen genes have sRNAs associated with them in pollen?
- Do the pollen genes have any other distinctive chromatin features in addition to TE-like methylation (histone mods, accessibility)?

Reviewer #3 (Remarks to the Author):

The manuscript entitled “Potent pollen gene regulation by DNA glycosylases in maize” by Zeng, Somers et al., reports the identification of genes under the control of active DNA demethylation in pollen. This active process is thought to be triggered by 2 DNA glycosylases: MDR1 or DNG102

Using microscopy, genetics and single cell transcriptomics the authors found a set of down-regulated genes (58) in double *mdr1 dng102* mutant pollen, suggesting that these DNA glycosylases act at these loci to release the expression of these particular genes.

Experiments are elegant and support the results.

While authors nicely show the specific expression of the candidate genes in bicellular microspore and in pollen I would suggest to better develop the relationship between expression profile and DNA methylation. This would strengthen the message of the story.

I would recommend to go further in the analysis:

Are these genes expressed under particular growth conditions, upon exposure to biotic/abiotic stresses?

Do these genes share similar genomic organization? number of introns? cis regulatory elements?

Where is DNA methylation predominantly located? Are there putative methylstats?

Figure 2D should be enlarged (zoom in) to better visualize the location of DNA methylation

In the results part I would suggest to give less details about the biological function of the gene products (“....When secreted into the apoplast, expansins and pectinases loosen cell walls.....”)

REVIEWER COMMENTS

Reviewer #1 (Remarks to the Author):

In this manuscript, Zeng and Somers et al performed single pollen RNA-seq using maize mutants heterozygous for two DNA demethylase genes. Because many genes that are normally methylated like TEs in somatic cells highly express in pollen-containing tissues, and double loss of function mutants of DNA demethylase genes show no transmission through pollen to next generation, the authors hypothesized that the TE-like methylated genes are demethylated and activated in pollen and those genes are important for pollen viability and function. The single pollen RNA-seq results clearly support the idea that DNA demethylase genes MDR1 and DNG102 are important for establishing pollen transcriptome. Single pollen RNA-seq results apparently show that double mutant pollens are different from single and WT pollens in overall transcriptome pattern and expression of cell wall related genes that tend to cluster in chromosomes are strongly downregulated. However, link of this downregulation with DNA methylation is not convincingly proved. For example, they can at least analyze DNA methylation in wild type pollen or pollen-containing tissue to see if the target genes (DEGs) are demethylated in pollen or not. In addition they can analyze DNA methylation pattern and/or DNA sequence features that may distinguish the genes that the DNA demethylases work on and genes/TEs that they do not, which may reveal important perspective on how DNA demethylases specifically target genes that should be activated in pollen. In conclusion, in the current form, I think the advancement of the understanding of the function of DNA demethylases in pollen function is limited.

To more directly link the regulation of potential DNG target genes with methylation, we prepared and analyzed DNA methylomes from pollen. As expected, the MPGs showed decreased methylation in pollen centered around TSSs (new Fig. 5A). The effect size was modest, but this may be explained because pollen is trinucleate, and only one of three nuclei (the vegetative nuclei) is expected to be demethylated. To confirm this idea, we first tried (unsuccessfully) to obtain purified cell type-specific nuclei (vegetative and sperm nuclei). As an alternative, we analyzed methylation levels in individual reads from pollen, as each read originates from a single DNA molecule but can report on multiple methylated bases. We found that individual methylation reads from pollen that overlapped the set of DNG target promoters showed a bimodal distribution, with most reads showing either high methylation or low methylation and few in between (revised Fig. 5C). More detailed analysis of five representative genes revealed that the ratio of unmethylated : methylated reads was roughly 1:2, as expected if the single vegetative nucleus was demethylated while the two sperm nuclei remained methylated (revised Supplemental Fig. 8).

As suggested by Reviewer 2, we also analyzed endosperm methylation data, since about half of pollen DNG targets are also demethylated in endosperm in Arabidopsis. This revealed demethylation of DNG targets in maize endosperm as well, even though their expression is pollen specific, which supports our model of a two-step activation requiring not just demethylation but also pollen specific factors to recruit high levels of RNA polymerase (Fig. 5 and Supplemental Fig. 8).

In terms of methylation patterns and structural features that distinguish DNG targets from other genes, we would like to point out that this has already been a major part of our analysis. In fact, we were able to identify DNG targets with high accuracy based on their methylation patterns, specifically high methylation in both CG and CHG contexts in their coding regions (TE-like methylation). This pattern is highly unusual for expressed, syntenically conserved genes with strong expression. We also described our finding that the majority of the DNG target genes are encoded with just one or two exons, which is unusual for expressed and syntenically conserved genes. Like other papers on DNG target genes in either endosperm or pollen, we did not include information on specific motifs associated with demethylation. Finding such motifs has proven to be difficult. We are still working on this problem, including using transgenes with DNG-target promoters, but this topic is beyond the scope of this paper.

Reviewer #2 (Remarks to the Author):

In this manuscript by Zeng et al, the authors uncover a fascinating class of genes that are highly expressed in maize pollen and are characterized by having TE-like methylation and also being down regulated (fail to switch on) in pollen in DNG mutants. The genes are enriched for functions related to cell wall modification and also dng mutants have pollen defects. This all supports a model whereby DNGs regulate the expression of genes critical for pollen function, presumably by demethylating them in pollen cells. This is a compelling story and points to an important function of the DNGs (although perhaps an indirect function, considering they do demethylate many other loci too), and the use of single pollen RNA-seq was an inspired approach. That said – and as acknowledged by the authors – pollen expression of methylated genes was previously reported in Arabidopsis, although this was investigated in less detail in the prior work. I have no doubt this is an important contribution and will be a highly cited paper in years to come. A couple of questions do come to mind that could make this a more complete discovery.

Major queries:

I would be itching to measure the demethylation of the genes in pollen cells to definitively establish that these loci are indeed demethylated in order to be expressed. This seems highly likely, but that data would be so helpful (I am not sure what the alternative scenario would be – demethylation of regulatory regions or a key TF? – but you never know!). Likely this is a technically challenging thing to do, probably requiring isolating VC nuclei at least in WT pollen. But this could also be discussed (eg in the discussion) as the TE-like methylation profiles are from leaves (?) and I assume a pollen cell methylome is not available for comparison. Or what about looking at methylation in the endosperm (as a proxy for DNG activity in the central cell), given that prior work in Arabidopsis and rice found that many loci demethylated in the vegetative nucleus were overlapped with loci demethylated in the central cell (eg Ibarra et al and Kim et al)?

We would like to thank the reviewer for the suggestion to use endosperm. As described in the response to Reviewer 1, this worked out very well, as did using whole pollen for demonstrating demethylation of these genes. See the last three paragraphs in the results section of the revised manuscript and Fig. 5 and Supplemental Fig. 8 for details.

A second limitation to note is the potential role of DNG105 in pollen. The expression pattern of DNG105 (Fig 3) combined with the stable expression of some of the “pollen methylated genes” in the pollen of the *dng* double mutant, does point to a role for DNG105, although without combining a *dng105* mutant with the other mutations, it is hard to say for sure and we can only speculate. Again, this is not something that can be easily done, and the authors do acknowledge this limitation.

The prospect of having to create double mutants with *dng105*, or more likely triple mutants, led us to focus our recent efforts on other experiments, including *mdr1* overexpression and searching for non-DNG factors in this pathway and for cis-regulatory elements. We did, however, obtain a mu insertion in *dng105* from the UniformMu collection just in case it would have a pollen fertility phenotype as a single mutant. It was fertile, so we set *dng105* aside for now.

A third query is whether these genes, and the process, is conserved in other species. For example, are the same genes (homologs) TE-methylated and pollen expressed in rice or sorghum? It would be even more compelling to see this in another species, or to determine if this is maize specific.

To assess whether the TE-like methylation patterns were shared with other species, we re-analyzed rice DNA methylation data focusing on expansin and polygalacturonase genes (two of the major families identified as MPGs in maize). These genes are already known to be strongly and specifically expressed in pollen in many species. Prior work by Scott Russell and colleagues has identified four clusters of these genes in rice, and we found that all four are made up primarily if not entirely of genes with TE-like methylation, as expected if they are activated by DNGs in pollen. We included a summary of this analysis in the revised results section and in Supplemental Figure 7.

Minor

- When referring to genes having TE-like methylation, it would be best to specify the tissue the methylation data is from “genes with TE-like methylation in leaf tissue”.

In the revised introduction, we have added “in leaf” to multiple occurrences of “TE-like methylation”.

- It is mentioned that there were 6 double mutant pollen grains sequenced, but there are 7 samples that cluster with double mutant transcriptome (Fig 1D)? Is it 6 or 7?

We were able to genotype 23 of the 26 pollen grains simply by linked SNPs. The other 3 were inferred by clustering, one of which clearly clustered with the 6 already-identified double mutants. We modified the relevant sentence in the results section to clarify: “Unsupervised hierarchical clustering of the single-pollen transcriptome data, including the three pollen transcriptomes with ambiguous genotypes, produced two distinct clusters, one with 19 pollen grains and one with 7 (Figure 1D)”.

- In Fig 2 A and B some axis numbers are cut off.

Thanks for pointing this out. We corrected the axis numbers in the revised Figure 2.

- Is it known if these pollen genes have sRNAs associated with them in pollen?

We have been examining small RNA data. The analysis is difficult to do in a definitive way because of ambiguities with mapping 21-25nt siRNAs to repetitive loci and variations in quantification depending on normalization methods. The short answer at this point is that DNG target genes have few siRNAs associated with them in leaf, early tassel, embryo, endosperm or pollen, as is the case for most functional genes. Consistent with this, DNG target genes have little CHH methylation (see Figure 5, Supplemental Figure 7, and Supplemental Figure 8). We still cannot rule out the possibility of a developmentally-regulated burst of siRNAs in the early embryo, for example, that could establish methylation at DNG target genes, and then be maintained without siRNAs throughout development. We are working on this, and we hope to publish another study on small RNAs in pollen and connections to chromatin, likely independent of DNGs.

- Do the pollen genes have any other distinctive chromatin features in addition to TE-like methylation (histone mods, accessibility)?

Ideally we would answer this question using histone modification and accessibility data from pollen, which we do not have now but hope to obtain in the future. As mentioned in the response to Reviewer 1, we have had trouble purifying nuclei from pollen, but this is a topic we would like to return to, hopefully to merge with our siRNAs in pollen study.

Reviewer #3 (Remarks to the Author):

The manuscript entitled “Potent pollen gene regulation by DNA glycosylases in maize” by Zeng, Somers et al., reports the identification of genes under the control of active DNA demethylation in pollen. This active process is thought to be triggered by 2 DNA glycosylases: MDR1 or DNG102

Using microscopy, genetics and single cell transcriptomics the authors found a set of down-regulated genes (58) in double *mdr1 dng102* mutant pollen, suggesting that these DNA glycosylases act at these loci to release the expression of these particular genes. Experiments are elegant and support the results.

While authors nicely show the specific expression of the candidate genes in bicellular microspore and in pollen I would suggest to better develop the relationship between expression profile and DNA methylation. This would strengthen the message of the story. I would recommend to go further in the analysis:

As recommended, and as described in the response to Reviewer 1 and 2, we have strengthened the connection between expression and DNA methylation through analysis of DNA methylation in pollen and endosperm. See the last three paragraphs in the results section of the revised manuscript and Fig. 5 and Supplemental Fig. 8 for details.

Are these genes expressed under particular growth conditions, upon exposure to biotic/abiotic stresses?

We have examined expression of DNG-target genes in ten tissues under normal developmental conditions, and only found strong expression in the pollen-containing tissues anther and tassel. It is an interesting idea to analyze expression under abnormal stress conditions, but we have not carried out such an analysis. We may pursue this in future work in conjunction with efforts to identify cis regulatory of DNG target genes.

Do these genes share similar genomic organization? number of introns? cis regulatory elements?

The genomic organization of DNA target genes is quite interesting, we think, and we describe in the results how half of the genes are encoded in genomic clusters, and more than two-thirds have only one or two introns. Like other papers on DNG target genes in either endosperm or pollen, we did not include information on specific motifs/CREs associated with demethylation. Finding such motifs has proven to be difficult. We are still working on this problem, including using transgenes with DNG-target promoters, but this topic is beyond the scope of this paper.

Where is DNA methylation predominantly located? Are there putative methylstats?

One of the surprising results of this study is that the DNA methylation is spread across the gene bodies of DNG targets, not just at the regulatory regions at each end. In fact, we defined TE-like methylation solely by methylation in coding DNA, although we found that it also occurs substantially in promoters and 3' ends. Figure 2D, Figure 5, Supplemental Figure 7, and Supplemental Figure 8 show this pattern. The demethylation in pollen was most evident in promoters. We have not done experiments to test for methylstats in the *mdr1* or *dng102* genes, though we would certainly like to know how DNG themselves are regulated.

Figure 2D should be enlarged (zoom in) to better visualize the location of DNA methylation

We wanted to keep Figure 2D zoomed out to illustrate how the gene copies can be separated by large and variable distances to avoid a misconception that these are simple head-to-tail tandem duplications. But we agree that showing a high-resolution image of single-base methylation

would be worthwhile, and we have done this for four example genes in the new Supplemental Figure 7.

In the results part I would suggest to give less details about the biological function of the gene products (“.....When secreted into the apoplast, expansins and pectinases loosen cell walls.....”)

While the mode of regulation of these genes is the focus of this manuscript, we think details on the biological function of the genes is worth including for two reasons. First, it is highly unusual for a gene regulatory mechanism to be enriched so strongly for so similar a set of genes (secreted cell wall factors). Second, the likely biological functions of these genes are of interest because of their relevance to poorly understood pollen biology and because they may have relevance to cell wall degradation for biofuels.

REVIEWER COMMENTS

Reviewer #1 (Remarks to the Author):

In the new manuscript my suggestions as well as other reviewers' suggestions were incorporated by conducting DNA methylation profiling in the WT pollen. The results show that the DNA methylation mainly in CG sites are reduced around TSS regions of MPGs in pollens compared to in embryo. This strengthens authors' conclusion that DNA methylation is involved in the regulation of MPGs. One further comment is the authors should show how significant the difference of DNA methylation is in Fig. 5A. Another comment is that they should discuss why only CG methylation is reduced and how the high expression is possible regardless of the still high level of CHG methylation in pollen. Furthermore, what the x-axis means in the new Supplementary Fig. 8 should be described in the legend. Or is there any other clearer way of showing the results such as violin plot?

Reviewer #2 (Remarks to the Author):

All my queries have been addressed, the new methylation data for pollen and endosperm significantly adds to the manuscript, this is great new data.

Minor suggestion:

- In new Fig 5 (a), it would be helpful to include leaf as well, for direct comparison. The legend in 5(b) could also be moved to the right of the panel so it does not overlap the line plot.

Reviewer #3 (Remarks to the Author):

The authors addressed all my comments and improved the manuscript.

Reviewer #1 (Remarks to the Author):

In the new manuscript my suggestions as well as other reviewers' suggestions were incorporated by conducting DNA methylation profiling in the WT pollen. The results show that the DNA methylation mainly in CG sites are reduced around TSS regions of MPGs in pollens compared to in embryo. This strengthens authors' conclusion that DNA methylation is involved in the regulation of MPGs. One further comment is the authors should show how significant the difference of DNA methylation is in Fig. 5A.

We have added a new Supplementary Figure 8 and the following paragraph to the Results section to demonstrate statistical significance:

“As an alternative way to quantify differences in mCG between tissues, we measured the average mCG value of the whole gene for each MPG and its 600-bp upstream region (promoter) (Supplementary Fig. 8). The mean per gene mCG value decreased 7% in pollen relative to embryo. For promoters it decreased 17% relative to embryo. While these decreases were small, they were statistically significant ($p = 0.0007$ for genes, $p = 0.001$ for promoters; one-tailed Wilcoxon signed rank test).”

Another comment is that they should discuss why only CG methylation is reduced and how the high expression is possible regardless of the still high level of CHG methylation in pollen.

Good point. We have added the following paragraph to the Results section as an explanation on why only CG methylation is reduced:

“CHG methylation (mCHG) appeared unaffected in MPGs in pollen. However, the baseline mCHG level in the rest of the genome was elevated in pollen relative to embryo (Fig. 5B). Given the context of a higher mCHG baseline in pollen relative to embryo, the similar mCHG level in MPGs could be consistent with partial CHG demethylation limited to one-third of nuclei in pollen. Weaker CHG demethylation compared to CG demethylation could be explained by either a preference of DNAs for mCG or by competing action of CHG methyltransferases adding mCHG in pollen.”

We suspect there is a strong but transient removal of all mCG and mCHG methylation before pollen maturation, but CHG methyltransferases begin putting some of the mCHG back. We would rather not speculate too much though, since we only have methylome data from mature pollen.

Furthermore, what the x-axis means in the new Supplementary Fig. 8 should be described in the legend. Or is there any other clearer way of showing the results such as violin plot?

The position on the X axis is randomly generated to prevent overlapping dots. We have added this to the legend (now Supplementary Fig. 9). We decided against a violin plot because the large number of dots with the exact value of 1 makes for hard-to-interpret violins. Our display also has the benefit of allowing inclusion of multiple biological replicates in a single distribution, but distinguishable by dot color.

Reviewer #2 (Remarks to the Author):

All my queries have been addressed, the new methylation data for pollen and endosperm significantly adds to the manuscript, this is great new data.

Minor suggestion:

- In new Fig 5 (a), it would be helpful to include leaf as well, for direct comparison.

We have added leaf, as well as tassel, in the new Supplemental Fig. 8, which is a more quantitative analysis of the same data as Fig. 5A.

The legend in 5(b) could also be moved to the right of the panel so it does not overlap the line plot.

Thanks for noticing that. We have moved the legend so it does not cover any lines.

Reviewer #3 (Remarks to the Author):

The authors addressed all my comments and improved the manuscript.